# Zero-Shot Video Grounding for Automatic Video Understanding in Sustainable Smart Cities

**Ping Wang** ⓘ, **Li Sun** ⓘ, **Liuan Wang** ⓘ and **Jun Sun** *ⓘ

Fujitsu Research & Development Center Co., Ltd., Beijing 100022, China
* Correspondence: sunjun@fujitsu.com

**Abstract:** Automatic video understanding is a crucial piece of technology which promotes urban sustainability. Video grounding is a fundamental component of video understanding that has been evolving quickly in recent years, but its use is restricted due to the high labeling costs and typical performance limitations imposed by the pre-defined training dataset. In this paper, a novel atom-based zero-shot video grounding (AZVG) method is proposed to retrieve the segments in the video that correspond to a given input sentence. Although it is training-free, the performance of AZVG is competitive to the weakly supervised methods and better than unsupervised SOTA methods on the Charades-STA dataset. The method can support flexible queries as well as different video content. It can play an important role in a wider range of urban living applications.

**Keywords:** video grounding; video understanding; zero-shot; multi-modality; CLIP

## 1. Introduction

Real-time perception of changes in the environment and the lives of residents is necessary to create a truly sustainable smart city. The growing popularity of surveillance cameras provides a solid guarantee of sustainable urban living. The development of social media has made people more willing to record and share their lives through videos. Technologies for automated video analysis are becoming more and more crucial in the face of massive amounts of data.

Temporal localization in untrimmed videos is a fundamental issue in video understanding. It has two subfields: temporal action localization [1] and video grounding [2]. Temporal action localization aims to find the start and end times as well as action labels in videos [3,4]. The actions are limited to pre-defined simple classes and find it challenging to cover complex scenarios in the real life. To overcome this limitation, video grounding was suggested in 2017 [5,6]. Video grounding is also called temporal sentence grounding or natural-language video localization. Natural language is used instead of pre-defined labels. Given a video and a sentence, it aims to find the time when an action in a sentence begins and ends in the video, as per the example shown in Figure 1. Video grounding is a more challenging task as it demands the ability not only to understand the video but also to break the modality gap between text and vision.

Video grounding aims to localize the start and end time of an event in a video based on the guidance of a given natural query. When processing massive amounts of video data, video grounding can greatly reduce manual annotation work. Breaking the barriers between video and text will increase the convenience of retrieval. In the field of video editing, it can assist creators in automatically localizing the content they need. In the field of security, it can help staff quickly locate abnormal behaviors that need attention from long-term surveillance videos. For video websites, it can also be used for precise timeline positioning of the searched content. In some scenarios, it is a better choice to replace manual audits with machines for privacy reasons.

Most methods for video grounding operate in a supervised manner. For each sentence, information on the start and end times in the video are needed. Traditional methods are

two-stage solutions following the rule of "propose and rank" [5,6]. In [7], a graph was used to explicitly model temporal relationships among proposals. Indeed, 2D-TAN [2] introduced a two-dimensional temporal map to model temporally adjacent relations of video clips and MS-2D-TAN [8] is the multi-scale version of 2D-TAN. To achieve more efficient performance in this task, end-to-end (one-stage) methods have become popular in recent years [9]. One-stage methods generate video clips related to the sentence directly. The authors of [10] proposed a method using an attention mechanism directly to predict the coordinates of the queried video clip. The authors of [9] used the distances between the frames within the ground-truth period and the start–end frames as dense supervisions to improve accuracy. With the development of reinforcement learning, SM-RL [11] used this technique in video grounding. The selection of start–end times could be regarded as a sequential decision-making process.

Query: person drinking from a cup.

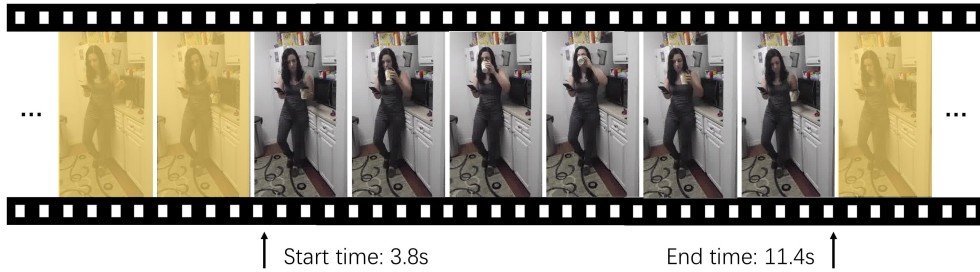

Start time: 3.8s          End time: 11.4s

**Figure 1.** An example of the video-grounding task. The inputs are the video and the query. The start and end times of the query in the video will be predicted.

Due to the huge cost of labeling, it is difficult to apply supervised methods to practical scenarios. In order to reduce the labeling cost, some weakly supervised methods [12–15] and even an unsupervised methods [16] are proposed. Weakly supervised methods require paired video-query knowledge without detailed segment annotations, whereas the unsupervised method requires only the video set and the query set for training [16]. Ref. [13] is the first work outlining a weakly supervised methods in an iterative way to obtain a temporal boundary prediction, and the segment is fed into the event captioner to generate a sentence. The authors of [12] added audio features to improve the performance. SCN [15] constructs a proposal generation module which can aggregate the context information to obtain candidate proposals. The authors of [14] trained a joint visual-text embedding and obtained the moment location using the latent alignment obtained by text-guided attention (TGA). DSCNet was proposed in 2022, and it is the first unsupervised method in the temporal video-grounding task [16]. DSCNet mines the deep semantic features from the query set to compose possible activities in each video. Compared with supervised methods, they have worse performance.

The supervised methods have high labeling costs as a large number of action moments and sentence pairs are needed. The weakly supervised and unsupervised methods can reduce the cost of data annotation to a certain extent, but data collection is still necessary. After collecting the data, training is needed for all of the supervised, weakly supervised, and unsupervised methods. Their performance depends heavily on the distribution of training data. As a result, it is hard to apply a model to another, different scenario and they cannot cope with out-of-domain cases. As real-life videos are very complex and changeable, the application of existing video ground methods is limited.

The challenges in data and generalization capabilities motivated us to propose a novel zero-shot video-grounding method by leveraging an off-the-shelf pre-trained model (e.g., CLIP [17]). The performance is competitive with the SOTA weakly supervised and unsupervised methods. In contrast to the previous methods, our solution has great value in real applications with obvious advantages. First, no labeling cost is needed as it is training-free. More importantly, it can overcome the limitation of training-set distribution.

As a result, it can be quickly applied to new scenes. The main contributions of our method are three-fold:

1. For candidate anchor generation (CAG), we propose the time-interval determinantal point process (TI-DPP) method. The anchors for the top-$n$ candidates should not only have high image-text similarity scores but also be mutually independent. Using TI-DPP, the top-$n$ candidate anchors will be recommended one-by-one in a greedy manner.
2. To obtain the precise moment, atom-based time period detection (ATPD) is proposed. This process includes two steps: splitting the video into atom actions and using a bi-directional search to merge the anchor atom regions with surrounding regions under various rules.
3. To enhance the robustness of the expression of the input query, prompting sentences generation (PSG) is proposed to select sentences that are accurate in meaning and diverse in description.

The rest of the article is organized as follows: Section 2 outlines the background and existing techniques related to this topic. Section 3 mainly describes the overall framework of the proposed method. Section 4 uses two popular datasets to verify the method. Section 5 is the conclusion.

## 2. Related Work

### 2.1. Multi-Modal Pretrained Models

In the last two years, the joint representation of images and text has attracted increasing attention due to its vast potential applications, e.g., CLIP [17], ALIGN [18], and WenLan [19]. They have provided a new paradigm for natural language processing and computer vision downstream tasks. Multi-modal pre-trained models are inspired by some previous works. In 2013, Socher trained image features to be closer to their associated words [20]. In 2017, Li collected images from the Internet and trained them to predict corresponding user comments [21]. These models could be used in many downstream tasks.

CLIP (Contrastive Language-Image Pre-training) was developed by OpenAI in 2021 [17], aimed at learning a joint representation of image and text. The pre-trained model could be used to estimate the semantic similarity between a sentence and an image. Trained by 400 million image–sentence pairs collected from the Internet, CLIP is a very powerful model which could be used in many computer vision tasks, such as image classification [17], object detection [22], image generation [23] and image manipulation [24]. For video-understanding tasks, CLIP is used in video-text retrieval [25] and video classification [26,27]. Some work uses the image and text embeddings provided by CLIP for further training [22,27], while some work [25] uses CLIP directly for the zero-shot purpose without training. CLIP has not yet been used in video grounding as it is difficult to perform temporal localization without training. This method provides a novel perspective from which to tackle these challenges.

### 2.2. Shot Boundary Detection

Shot boundary detection (SBD) is also called shot transition detection. The task aims to detect the position of frames where the shot changes [28]. There are three main steps in this task: obtaining representative visual features, measuring similarity and transition detection [29]. The authors of [30] selected candidate segments using adaptive thresholds and used singular value decomposition (SVD) to increase the speed. The authors of [29] employed orthogonal polynomials to represent the visual information. At present, there are also some methods based on deep learning. The authors of [31] constructed a new pipeline based on features extracted from CNN. TSSBD [28] is a two-stage method which can, firstly, distinguish abrupt shots and then locate gradual shot changes using a 3D convolutional neural network.

Video grounding is a multi-modal task which tries to locate video periods based on text descriptions while shot boundary detection is only in the visual modality. The difference

between "shot boundary" in SBD and "time periods" in video grounding is obvious: shot boundary describes the transition between consecutive shots, whereas time periods in the video-grounding task might involve several consecutive actions as the text description may be very complex or abstract. Our ATPD module has something in common with shot boundary detection. Before locating moments related to text description, we split the video into atom actions just as shot boundary detection does. However, our method is not sensitive to the action segmentation performance, as the process of bi-directional merging will follow and the metrics in video grounding allow for some deviations in boundaries.

### 2.3. Text Augmentation

Text augmentation is important for some NLP (natural language processing) tasks. It can generate new data for small datasets and balance imbalanced classes. There are various methods of text augmentation, from data space to feature space [32]. Adding noise and synonym replacement are popular forms, both being simple and effective. At the phrase level, structure-based transformation [33] and interpolation [34] are generally used. At the document level, back-translation [35] and generative models [36] are popular.

Back translation is a widely used NLP augmentation method [35]. Back translation means translating the sentence to another language and then translating the result back to the original language. Figure 2 shows an example of back translation. The sentence after the back translation will have the same meaning as the original sentence, albeit in a different written format.

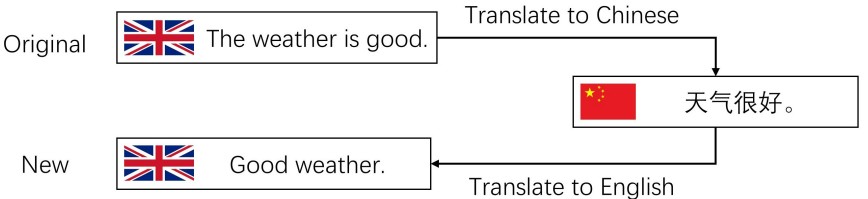

**Figure 2.** An example of back translation.

Previous work in video grounding omits the effort from the text end, and only original sentences are used in the training and testing process. Herein, we introduce a new perspective to improve the performance. A task named prompting sentences generation (PSG) is presented. Unlike text augmentation in some NLP tasks, our purpose is not to make the training process more robust, but to improve the matching performance for the input query and the video frames. As a result, we do not choose to generate new sentences by adding noise. Back translation is a suitable choice for our purpose. In addition, "prompt" [37,38] is a hot topic in NLP, whereas "prompting sentences" in this work has a very different meaning. The usage of "prompting sentences" is more like ensembling and it is training-free.

In order to generate back-translation sentences, we use the Google Translation API. It supports 109 different languages, including English and 108 other languages. For each original sentence, 108 back-translation results can be obtained, and they can be regarded as a source pool. In contrast to NLP augmentation, we have a much stricter demand for the generated sentences that will participate in the decision-making process. We want the prompting sentences to be accurate in meaning but diverse in description. Our method of PSG studies how to select better sentence groups for ensembling.

### 3. Materials and Methods

The pipeline of our method is shown in Figure 3. We begin with a straightforward approach to using the pre-trained image-text model, for example, CLIP. Using CLIP, the text feature of the sentence and an image feature list for the frames can be obtained. If one frame is more related to the sentence, the cosine similarity score of the text feature and the frame feature will be higher. Different from other tasks such as video retrieval, video

grounding aims to find the temporal location of a sentence happening in the video. Based thereon, it is reasonable to start by finding time anchors with high possibilities for the event. After obtaining the representative frames as coarse localization anchors, the method will expand the results to time ranges. In AZVG, the video-grounding task is divided into sub-problems: finding anchor frames that are most related to the sentence, segmenting a video to atom actions, and merging adjacent regions based on certain rules.

As shown in Figure 3, the input is a video and a sentence. The video will be decoded into frames and then image features will be extracted for each frame. For the text end, the original sentence and sentences generated from the PSG (prompting sentences generation) module will be used. As shown in Figure 3, prompting sentences will be generated based on the original sentence "a person looks at a book". The new sentences will have the same meaning as the original one, but with diverse discription. After extracting the text feature of the sentences, a frame–text similarity curve can be obtained. Using this curve, candidate anchors will be proposed in the CAG module (candidate anchors generation). The anchors will have high image–text similarity and action independency. After that, for each anchor, a concise period will be generated using the ATPD module (atom-based time-period detection). Finally, we obtain the top-*n* candidate periods for the given sentence. Each module in Figure 3 will be introduced in this section.

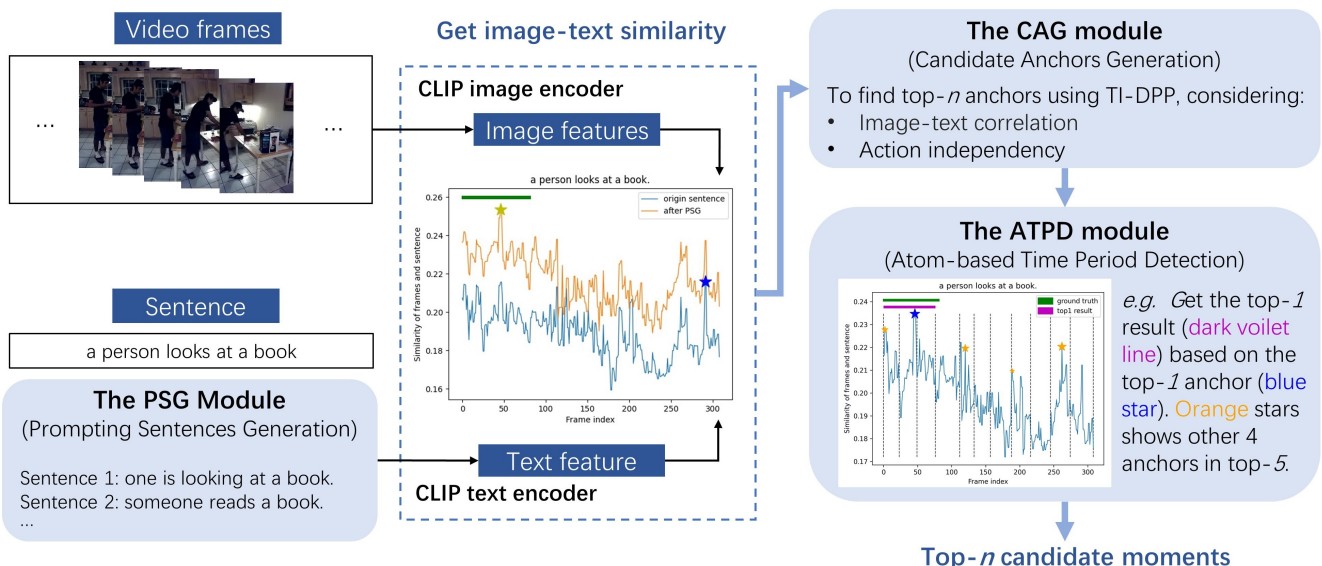

**Figure 3.** The pipeline of the AZVG method. Image features of video frames will be extracted using the CLIP image encoder. The text feature of the original sentence or prompting sentences will be extracted using the CLIP text encoder. A similarity curve of text and frames will be computed. The CAG module will generate top-*n* candidate anchors, and the anchors will be expanded to concise moments after the ATPD module.

### 3.1. Candidate Anchors Generation

The top-*n* performance is an important part of the evaluation metrics of video grounding. As the output of this task is a time period, how can the top-*n* results be obtained? After computing the similarity of the sentence (via features from the text encoder in CLIP) and each frame (via features from the image encoder in CLIP, 10 fps), a frame–text similarity curve is plotted. There is an observation that the peak of the frame–text similarity curve is usually among the ground-truth time region. However, the start and end times in the ground truth are hard to obtain only by the frame–text similarity. That is to say, the curve can be used to make the coarse localization whereas it is not good at obtaining the time-range information. As a result, our strategy is to find *n* representative anchors

first and then expand them to $n$ time periods based on atom action segmentation. The candidate-anchors-generation (CAG) module is for $n$ representative frames, or anchors.

Intuitively, the top-$n$ anchors should not be $n$ highest peaks in the similarity curve, as moments rather than time points are demanded in this task. If simply selecting $n$ highest global peaks, there is a high probability that they are in the same atom action. That means that if there is more than one event happening in the video, these events should be independent of each other. Therefore, any two possible candidate anchors should not be too close. It is better to find the top-$n$ anchors in local peaks which are relatively distant from each other in the time axis. Considering the physical meaning, there are two factors to be considered in selecting the anchors: the image–text similarity score and time intervals among the candidates.

To optimize the time interval and the text–image similarity score at the same time, a method named TI-DPP (time interval determinantal point process) is proposed. TI-DPP is inspired by the determinant point process (DPP), a method for subset selection. DPP is a probabilistic model which can express diversity among elements in a set [39]. It was first proposed to ascertain fermion system distributions at thermal equilibrium [40] and is currently widely used in various tasks, such as content summarization, image searching, and recommendation systems, as it can find results of high relevance and diversity [41].

As an elegant probabilistic model, DPP has the ability to express negative interactions. A point process $\mathcal{P}$ on a ground set $\mathcal{Y}$ is a probability measure to obtain finite subsets of $\mathcal{Y}$ [39]. In the discrete case, a point process $\mathcal{P}$ on the set $\mathcal{Y} = \{1, 2, ..., M\}$ is a probability measure on the set of all subsets of $\mathcal{Y}$. We call $\mathcal{P}$ a determinantal point process when $Y$ is a random subset drawn according to $\mathcal{P}$ and for every subset $A$,

$$\mathcal{P}(A \subseteq \mathcal{Y}) = det(\mathbf{L}_A) \tag{1}$$

$\mathbf{L}$ is a real, positive semidefinite (PSD) kernel matrix with the size of $M \times M$. $\mathbf{L}_A$ denotes the restriction of $\mathbf{L}$ to the entries indexed by elements of $A$. Normalization is not needed. [39]. We refer to $\mathbf{L}$ as the marginal kernel since it contains all the information needed to compute the probability of any subset $A$ being included in $\mathcal{Y}$ [39]. Under many cases, we want to add a cardinality constraint on $\mathcal{Y}$ to return a subset of fixed size with the highest probability.

The target of TI-DPP is to find a subset $\mathcal{C}$ of anchor frames with high text–image similarity scores and longer time intervals between each other. The problem of finding anchors for top-$n$ candidates can be modeled as obtaining the subset with $n$ elements. A video can be regarded as a frame set $\mathcal{H}$ comprising $N$ frames. To guarantee the time-interval constraints, a linear time-interval matrix $\mathbf{D}$ is designed. The size of $\mathbf{D}$ is $N \times N$. Element $\mathbf{D}_{ij}$ is determined by the time interval between frame $i$ and frame $j$. $\mathbf{D}_{ij}$ is in a negative correlation with the time interval between $i$ and $j$.

$$\mathbf{D}_{ij} = 1 - \ell \times |i - j| \tag{2}$$

$$\ell = \frac{1}{N-1}, \quad N > 1 \tag{3}$$

where $\ell$ denotes the time step, which is related to $N$, the number of video frames. According to the function, when computing the distance of the image and itself ($i$ equals $j$), $\mathbf{D}_{ij}$ is 1. When computing the distance of the first frame and the last frame, $\mathbf{D}_{ij}$ is 0.

Besides the time interval matrix $\mathbf{D}$, $r$ is defined as the frame–text similarity score. In [41], a kernel matrix is constructed to describe the diversity and relevance. In TI-DPP, a kernel matrix $\mathbf{L}$ is designed for the defined time-interval matrix and frame–text similarity score.

$$\mathbf{L} = Diag(r) \cdot \mathbf{D} \cdot Diag(r) \tag{4}$$

where $Diag(r)$ is a diagonal matrix, whose diagonal vector is $r$.

The kernel matrix $\mathbf{L}$ is indexed by its frame indexes.. After obtaining the kernel matrix, our target is to find a subset $\mathcal{C}$ from $\mathbf{L}$:

$$\mathcal{C} \leftarrow argmax_{\mathcal{C} \subseteq \mathcal{H}} det(\mathbf{L}) \tag{5}$$

It is a maximum a posteriori (MAP) inference process to obtain the subset $\mathcal{C}$ which has the largest determinant among all subsets of $\mathbf{L}$. A greedy computing method is used, similar to DPP [41]. The top result will be the frame with the largest frame–text similarity score. Figure 4 shows an example of generated anchors which are independent local maxima.

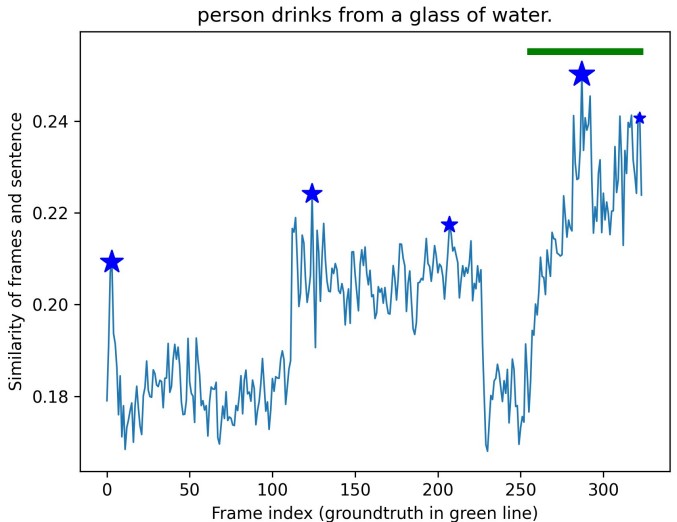

**Figure 4.** An example of peaks for top 5. The blue curve shows the frame–text similarity of frames. The green line shows ground truth of the time period for the sentence. The blue stars are the selected anchors for top-5 candidates and the largest one is for the top result.

### 3.2. Atom-Based Time Period Detection (ATPD)

To solve the challenge of obtaining start and end times without training, atom-based time period detection (ATPD) is introduced. This process mainly has two steps: splitting the video into atom action regions and using bi-directional search to combine the anchor atom regions with surrounding regions under certain rules. An atom action is an action that can no longer be divided. Frames in the same atom action are more likely to be similar. Intuitively, the atom actions of the video can be determined based on the self-similarity matrix of all frames. After the segmentation of atom actions, anchors for the top-$n$ candidates can be used to obtain the final result. For each anchor, the time period estimation will start from the atom region to which the peak belongs and expand to surrounding regions in two directions based on the frame–text similarity curve.

Figure 5 shows an example of a self-similarity matrix of frame features. The frame features are extracted by the image encoder of CLIP, then the cosine similarity of each two frames can be calculated to obtain the final self-similarity matrix. Some dark blue blocks on the diagonal of the matrix can be observed: these denote a higher similarity, and they can be regarded as belonging to the same atom action.

Inspired by the music segmentation literature [42], a Gaussian checkerboard kernel is used to obtain atom actions. The Gaussian checkerboard kernel combines the Gaussian function and the checkerboard kernel by multiplication. The checkerboard kernel is a special kernel which can be decomposed into "coherence" and "anti-coherence". An example of a 5 × 5 checkerboard kernel can be written as:

$$\mathbf{K}_{Checkerboard} = \begin{bmatrix} -1 & -1 & 0 & 1 & 1 \\ -1 & -1 & 0 & 1 & 1 \\ 0 & 0 & 0 & 0 & 0 \\ 1 & 1 & 0 & -1 & -1 \\ 1 & 1 & 0 & -1 & -1 \end{bmatrix} \tag{6}$$

After filtering the self-similarity matrix with the Gaussian checkerboard kernel, a function of normalization will be performed. We undertake element-wise product for the kernel and frame–frame feature similarity matrix and then sum the results of each column to obtain the output curve. Figure 6 shows an example of the output curve of added results after the Gaussian checkerboard filtering. Peaks in the curve will be used to split the whole video into atom actions: set **A** $\{a_1, a_2,...,a_\kappa\}$, where $\kappa$ is the number of atom actions, as shown in Figure 7.

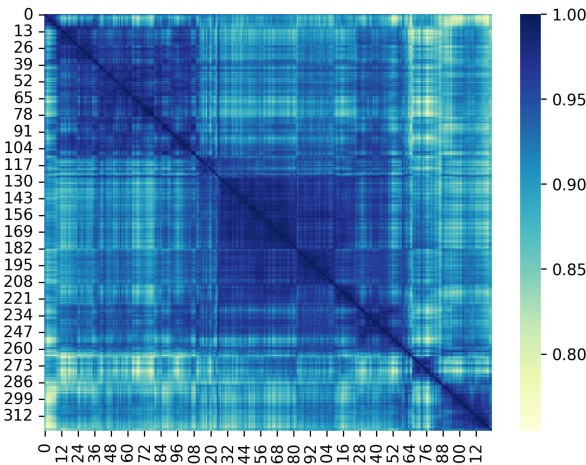

**Figure 5.** Example of self-similarity matrix of image features. The x-axis coordinates and the y-axis coordinates show the frame index. Each element in the matrix is the similarity of CLIP features from two frames. Some blocks in the diagnostics direction can be observed.

In Section 3.1, the top-$n$ anchors are obtained from the frame–text similarity curve. Using the anchors and the atom action segmentation result, the concise time period for the sentence can be obtained. One single atom action region is not enough to cover the sentence, as an event can generally be divided into several sub-events. We solve this problem by searching neighbors of the atom action region $a_l$ where the anchor exists from two directions: forward and backward.

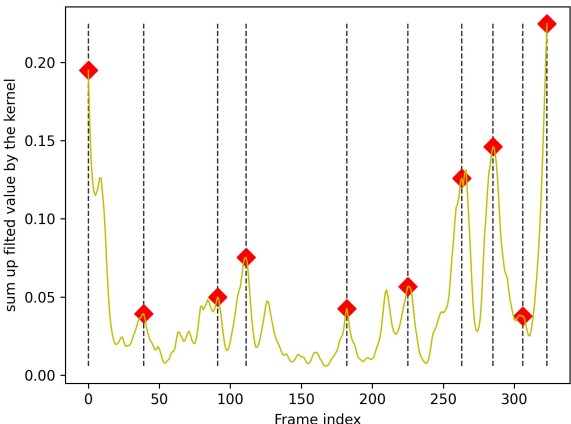

**Figure 6.** Marks for the segmentation.

In the bi-directional search process, three factors are mainly considered to determine whether the current atom action period $a_\varepsilon$ should be added to the final result or the search should be stopped: $p(a_\varepsilon)$ describes the peak value of the frame–text similarity curve for this atom action, $f(a_\varepsilon)$ describes the frame–text similarity flatness for this atom action and $\omega(a_\varepsilon)$ describes the compact score for this atom action. The final score $\mu(a_\varepsilon)$ equals $p(a_\varepsilon) \cdot f(a_\varepsilon) \cdot \omega(a_\varepsilon)$. $\varepsilon$ is the index of the current atom region, and $l$ is the index of the

anchor atom region. For each frame $i$, let $s(i)$ be the text–image similarity score, and $\bar{s}(a_\varepsilon)$ be the average frame–text similarity score for an action period.

$$p(a_\varepsilon) = \frac{\max\limits_{i \in a_\varepsilon} s(i)}{\max\limits_{i \in \mathbf{A}} s(i)} \tag{7}$$

$$f(a_\varepsilon) = \frac{\bar{s}(a_\varepsilon)}{\bar{s}(\mathbf{A})} \tag{8}$$

$$\omega(a_\varepsilon) = e^{-|\varepsilon - l|} \tag{9}$$

$$\mu(a_\varepsilon) = p(a_\varepsilon) \cdot f(a_\varepsilon) \cdot \omega(a_\varepsilon) \tag{10}$$

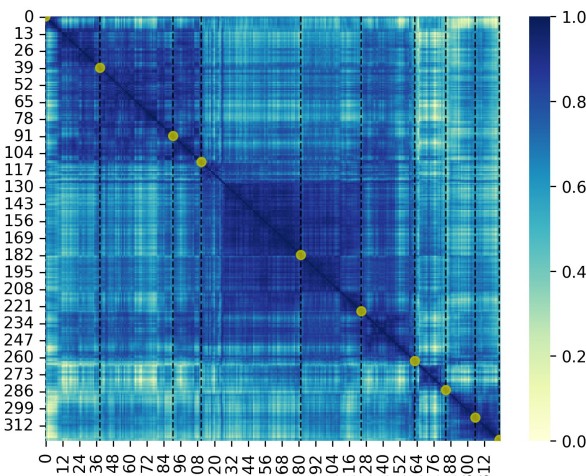

**Figure 7.** The atom actions.

The searching process will start from the atom action region where the current anchor is. The value of $\mu(a_\varepsilon)$ for each atom action period will be computed before the searching process stops. Details of the bi-directional searching process for an anchor and its corresponding atom region are shown in Algorithm 1. To obtain the top-*n* candidates, the process of ATPD will be repeated *n* times for each anchor generated by TI-DPP. Figure 8 shows an example of obtaining the top time period result.

### 3.3. Prompting Sentences Generation (PSG)

Text augmentation is widely used in natural language processing. However, in the video-grounding task, little attention has been paid to the text end. In this work, a new concept of "prompting sentences generation (PSG)" is raised. It is very different from the text augmentation used in NLP tasks. In NLP tasks, text augmentation is used to train a more robust model. It is generally performed by introducing some data noise, such as removing (or adding) words, replacing synonyms, and so on. Prompting sentences are used to improve the final decision in video grounding. To maximize the potential of CLIP, accurate but diverse prompting sentences are required. The use of prompting sentences is closer to "ensemble" rather than "augmentation". We want the prompting sentences to be diverse in description, whereas all of them have a high semantic similarity with the original sentence. In this article, back translation is adopted to generate plenty of sentences, and a subset of sentences satisfying "exactness" and "diversity" will be selected. In this work, two methods are presented to select the subset of prompting sentences from different perspectives: sentence level and language level. In addition, an appropriate metric is designed to evaluate the outlook similarity of two sentences, to guarantee the diversity in description.

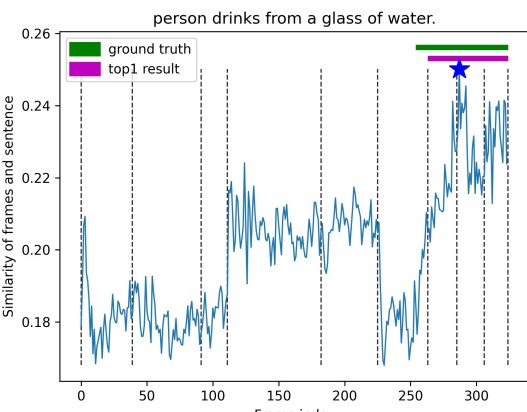

**Figure 8.** An example of the top time period result. The dotted lines show the segmentation result of atom actions. The blue star is the top anchor. The green line shows the ground truth and the dark violet line shows the top time period result.

There are $k$ query sentences in the sentence set **O**: $\{o_1, o_2, o_3, ...o_k\}$. Each sentence corresponds to a time period in the video. The Google translation API, which supports English and other 108 languages, is used to perform back translation for the sentences. They can be regarded as a back-translation function set $\psi$ with 108 functions for different languages: $\{\phi_1, \phi_2, \phi_3, ...\phi_{108}\}$. For example, the original sentence $o_\tau$ utilizes the back-translation functions to obtain 108 new sentences in $\psi(o_\tau)$: $\{\phi_1(o_\tau), \phi_2(o_\tau), \phi_3(o_\tau), ...\phi_{108}(o_\tau)\}$. The target is to find $c$ best sentences of both "exactness" and "diversity" from the generated sentence set $\{\phi_1(o_\tau), \phi_2(o_\tau), \phi_3(o_\tau), ...\phi_{108}(o_\tau)\}$. It will be discussed from two perspectives: language level and sentence level.

In the language-level prompting sentences generation, $c$ back-translation functions of different languages will be selected from the function set $\psi$. In sentence-level prompting sentences generation, for each sentence, $c$ new sentences will be selected from $\psi(o_\tau)$. In both perspectives, "exactness" and "diversity" are considered. To guarantee "exactness", a pre-trained semantic matching model [43] will be used to evaluate whether the new sentence has the same meaning as the original sentence. To guarantee "diversity", a new metric to evaluate the outlook similarity of two sentences is designed.

After obtaining $c$ prompting sentences of correctness and variety, they will be used together with the original sentence. Two ways are explored to take advantage of the prompting sentences for better performance: averaging and middling. Averaging means obtaining $c + 1$ frame–text similarity curves between frames and these $c + 1$ sentences and then averaging the curves to obtain a more robust new curve to replace the original one. Middling means after obtaining the frame–text similarity curves of the $c + 1$ sentences, the curve whose score peak is located in the middle of the $c + 1$ score peak locations is chosen. Figure 9 shows an example of how the PSG module works. As shown in the figure, the maximum peak in original sentence is not among the ground truth. However, using the prompting sentences, the maximum peak is among the ground truth.

### 3.3.1. Semantic Matching

As the selected prompting sentences should be exact and diverse, semantic matching is used to guarantee the exactness. Not all back-translation sentences have the correct meaning, and there are usually some low-quality sentences that mislead as to the original meaning. A pre-trained semantic matching model is deployed to solve this problem by providing a similarity score for the original sentence and each sentence generated through back translation. MPNet [43] with pre-trained weights is chosen as a semantic matching filter to remove some low-quality sentences or languages. Let $Se$ denote the semantic matching function; for sentence $o_\tau$, the semantic similarity score of the original sentence and a back-translation sentence $\phi_l(o_\tau)$ can be written as $Se(o_\tau, \phi_l(o_\tau))$.

---

**Algorithm 1** Bi-directional search for time period

---

**Input:**

1: Atom action result **A**.

2: Atom region $a_l$, to which the current anchor belongs.

3: Threshold $\delta$ for eligible atom regions.

**Output:** Time periods of the combined selected atom actions.

4:

5: Initially put $a_l$ in the result list, let current atom index $\xi = l$.

6:

7: % Processing: forward search

8: **if** $\xi < \kappa$ **then**

9:      **repeat**

10:          $\xi = \xi + 1$

11:          Get $p(a_\xi), f(a_\xi), \omega(a_\xi)$ according to (7) (8) (9).

12:          Compute $\mu(a_\xi) = p(a_\xi) \cdot f(a_\xi) \cdot \omega(a_\xi)$

13:          **if** $\mu(a_\xi) \geq \delta$ **then**

14:             Put current action region $a_\xi$ in the result list.

15:          **end if**

16:      **until** $\mu(a_\xi) < \delta$

17: **end if**

18:

19: % Processing: backward search

20: **if** $\xi > 0$ **then**

21:      **repeat**

22:          $\xi = \xi - 1$

23:          Get $p(a_\xi), f(a_\xi), \omega(a_\xi)$ according to (7) (8) (9).

24:          Compute $\mu(a_\xi) = p(a_\xi) \cdot f(a_\xi) \cdot \omega(a_\xi)$

25:          **if** $\mu(a_\xi) \geq \delta$ **then**

26:             Put current action region $a_\xi$ in the result list.

27:          **end if**

28:      **until** $\mu(a_\xi) < \delta$

29: **end if**

30: **return** The combined result list for atom action periods.

---

3.3.2. Evaluating the Outlook Similarity of Two Sentences

In this paper, a new metric called Word Set IoU (Intersection over Union) is established to evaluate the outlook similarity of two sentences, which can better fit the CLIP model. As CLIP is robust for a sentence with the order of words changed [17], the unordered word set in lowercase is chosen to represent a sentence. Sentence $\phi_i(o_\tau)$ and sentence $\phi_j(o_\tau)$ are back-translation results from language $i$ and language $j$ for the original sentence $o_\tau$. They can be represented by two unordered word sets $W(\phi_i(o_\tau))$ and $W(\phi_j(o_\tau))$. $W$ is the function to change a sentence to an unordered word set in lowercase. In addition. the similarity computed by Word Set IoU is:

$$S_o(\phi_i(o_\tau), \phi_j(o_\tau)) = \frac{Intersect(W(\phi_i(o_\tau)), W(\phi_j(o_\tau)))}{Union(W(\phi_i(o_\tau)), W(\phi_j(o_\tau)))} \tag{11}$$

$Intersect(W(\phi_i(o_\tau)), W(\phi_j(o_\tau))$ means the intersection of the two word sets, or to say the number of same words in the two sentences. $Union(W(\phi_i(o_\tau)), W(\phi_j(o_\tau))$ means the union of the two word sets, or the number of all different words in the two sentences. For example:

- Sentence $\phi_i(o_\tau)$: "A man was standing in the bathroom holding glasses" can be separated into the unordered set $W(\phi_i(o_\tau))$: (a, man, was, standing, in, the, bathroom, holding, glasses).

- Sentence $\phi_j(o_\tau)$: "a person is standing in the bathroom holding a glass" can be separated into the unordered set $W(\phi_j(o_\tau))$: (a, person, is, standing, in, the, bathroom, holding, a, glass).

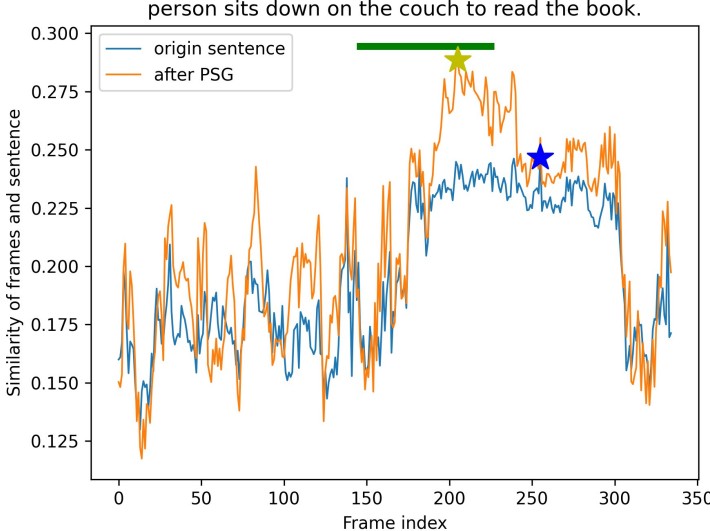

**Figure 9.** An example showing how PSG works. The blue curve shows the similarity of frames and the original sentence. The yellow curve is generated by averaging $c+1$ similarity curves of frames and the $c+1$ sentences. The $c+1$ sentences include the original sentence and $c$ sentences generated by the language-level PSG.

The intersection count in $W(\phi_i(o_\tau))$ and $W(\phi_j(o_\tau))$ is 6. The union count in $W(\phi_i(o_\tau))$ and $W(\phi_j(o_\tau))$ is 11. Therefore, the sentence similarity $S_o(\phi_i(o_\tau), \phi_j(o_\tau))$ is 6/11. The newly designed metric to evaluate the similarity of two sentences will be used to guarantee the diversity of selected sentences or selected languages.

### 3.3.3. Language-Level PSG

The language-level prompting sentences generation will select $c$ languages meeting the exactness requirements and with the largest diversity among all languages. The method mainly includes several steps: filtering some languages of low translation quality, constructing the language-level similarity matrix, and obtaining a subset of the languages to maximize the diversity.

Firstly, the semantic matching scores are calculated for sentences generated by the function set $\psi$. For each language, the average semantic matching score $\Gamma_i$ can be computed as follows:

$$\Gamma_i = \frac{\sum_{\tau=1}^{k} Se(o_\tau, \phi_i(o_\tau))}{k} \tag{12}$$

For the 108 languages, the average semantic scores are computed:$\{\Gamma_1, \Gamma_2, ..., \Gamma_{108}\}$. The $\sigma$ languages with the highest average semantic scores are selected.

Then the language-level outlook similarity matrix $\mathbf{L}$ is constructed. The language-level outlook similarity matrix represents the outlook similarity of results from every two languages. Currently, there are $\sigma$ languages left after the semantic matching filtering; therefore, $\mathbf{L}$ is of measure $(\sigma, \sigma)$. Element $\mathbf{L}_{ij}$ means the outlook similarity of language $i$ and language $j$. For each of the $k$ sentences in the entire dataset, the similarity score of the sentence from language $i$ and language $j$ is calculated as follows.

$$\mathbf{L}_{ij} = \frac{\sum_{\tau=1}^{k} S_o(\phi_i(o_\tau), \phi_j(o_\tau))}{k} \tag{13}$$

After $\mathbf{L}$ is obtained, the $c \times c$ submatrix of $\mathbf{L}$ with the largest determinant is calculated. This step can be regarded as a simplified process of DPP since only diversity is regarded

here. The $c \times c$ submatrix points to $c$ selected languages with maximum diversity in outlooks. Algorithm 2 will show the details.

---

**Algorithm 2** Language-level PSG

---

**Input:**
1: Back-translation functions $\boldsymbol{\psi}$: $\{\phi_1, \phi_2, \phi_3, ...\phi_{108}\}$
2: Sentence set **O**: $\{o_1, o_2, o_3, ...o_k\}$
**Output:** $c$ prompting sentences for each original sentence. $\leftarrow$ Selected $c$ languages
3: Initially obtain back-translation results: $\boldsymbol{\psi}(\mathbf{O})$
4: **for** $\phi_i$ in $\boldsymbol{\psi}$ **do**
5:     Compute the average semantic matching score $\Gamma_i$ according to (12).
6: **end for**
7: Select $\sigma$ languages by average semantic matching scores.
8:
9: Construct the language-level similarity matrix **L** for the left $\sigma$ languages:
10: **for** $\phi_i, \phi_j$ in $\sigma$ languages **do**
11:     Compute $\mathbf{L}_{ij}$ according to (13).
12: **end for**
13: Calculate $c \times c$ submatrix of **L** of largest determinant.
14: Get $c$ languages from the submatrix.
15:
16: **for** $o_\tau$ in **O** **do**
17:     Compute prompting sentences using the $c$ languages.
18: **end for**

---

Table 1 shows an example of language-level prompting sentence generation. Ten sentences are generated by back-translation functions of selected ten languages. As the language subset computing process is in a greedy manner, the first selected language can generate sentences with the largest difference from the original sentences. From a language-level point of view, the selected languages are in the highest diversity. However, language-level PSG cannot guarantee the maximized prompting sentence diversity for each original sentence. The reason is that the method will select languages rather than sentences. For example, in Table 1, sentence 7 is the same as sentence 10. Though the language for sentence 7 and language for sentence 10 can generate relatively different sentences based on statistics, it does not work for this sentence.

3.3.4. Sentence-Level PSG

In the sentence-level method, $c$ sentences of diversity and exactness will be selected for each original sentence. Differing from the language-level method, every sentence will repeat the selection process. For each sentence $o_\tau$, the steps are as follows:

Firstly, for language back-translation function $\phi_i$ from the function set $\boldsymbol{\psi}$, the semantic matching score of the original sentence $o_\tau$ is computed with the back-translation result sentence $\phi_i(o_\tau)$: $Se(o_\tau, \phi_i(o_\tau))$. There are a total of 108 back-translation functions of different languages in $\boldsymbol{\psi}$, so 108 semantic scores can be obtained. The $\sigma$ prompting sentences with good semantic matching performance are selected.

Then, the sentence-level outlook self-similarity matrix $\mathbf{L}_s(o_\tau, \boldsymbol{\psi})$ for sentence $o_\tau$ is constructed. Differing from the language-level method with only one outlook similarity matrix, in the sentence-level method, each sentence will have an outlook similarity matrix. Currently, there are $\sigma$ prompting sentences left for $o_\tau$, shape of $\mathbf{L}_s(o_\tau, \boldsymbol{\psi})$ is $(\sigma, \sigma)$. Element $\mathbf{L}_s(o_\tau, \boldsymbol{\psi})_{ij}$ denotes the outlook similarity of sentences from language $i$ and language $j$:

$$\mathbf{L}_s(o_\tau, \boldsymbol{\psi})_{ij} = S_o(\phi_i(o_\tau), \phi_j(o_\tau)) \tag{14}$$

**Table 1.** Ten examples of language-level prompting generation

| Type | Sentence |
|------|----------|
| Original sentence | a man stands in the bathroom holding a glass. |
| Generated sentence 1 | A man is holding a glass in the bathroom. |
| Generated sentence 2 | a man was standing in the bathroom holding glass. |
| Generated sentence 3 | A person is standing in the bathroom holding a glass |
| Generated sentence 4 | a man is standing in the bathroom holding a glass. |
| Generated sentence 5 | someone is standing in the bathroom holding a glass. |
| Generated sentence 6 | a man is standing in the bathroom with a glass. |
| Generated sentence 7 | Someone standing in the bathroom holding a glass. |
| Generated sentence 8 | the man is standing in the bathroom with a bottle. |
| Generated sentence 9 | Someone stands in the bathroom holding a glass. |
| Generated sentence 10 | someone standing in the bathroom holding a glass. |

Finally, the $c \times c$ submatrix of $\mathbf{L}_s(o_\tau, \boldsymbol{\psi})$ with the largest determinant is computed.

Algorithm 3 shows the details of sentence-level PSG. For each original sentence, we can select $c$ best prompting sentences (from 108 sentences) which have both diversity as well as exactness after these steps.

---

**Algorithm 3** Sentence-level PSG

---

**Input:**

1: Back-translation functions $\boldsymbol{\psi}$: $\{\phi_1, \phi_2, \phi_3, ... \phi_{108}\}$
2: Sentence set **O**: $\{o_1, o_2, o_3, ... o_k\}$

**Output:** $c$ prompting sentences for each original sentence.

3:
4: **for** $o_\tau$ in **O** **do**
5:     Get semantic matching scores $Se(o_\tau, \boldsymbol{\psi}(o_\tau))$ for $o_\tau$ and the back-translation results $\boldsymbol{\psi}(o_\tau)$.
6:     Select $\sigma$ sentences by semantic matching scores.
7:
8:     Construct the outlook similarity matrix $\mathbf{L}_s(o_\tau, \boldsymbol{\psi}_\sigma)$ for the $\sigma$ sentences.
9:     Compute $c \times c$ submatrix of $\mathbf{L}_s(o_\tau, \boldsymbol{\psi}_\sigma)$ with the largest determinant.
      **return** $c$ prompting sentences for $o_\tau$.
10: **end for**

---

Table 2 shows an example of sentence-level prompting sentences generation. It can be observed that, compared with the language-level result, the sentence-level result is better in diversity whereas worse in exactness. That is because a subset of the most diverse sentences will be selected for each sentence in the sentence-level method. The wrong sentences which fail to be filtered by semantic matching usually look different from other sentences so they are easier to pick. For example, in Table 2, sentence 6 is a wrong sentence.

**Table 2.** An example of sentence-level prompting generation.

| Type | Sentence |
|------|----------|
| Original sentence | a man stands in the bathroom holding a glass. |
| Generated sentence 1 | a person stands in the bathroom holding a glass. |
| Generated sentence 2 | the man is standing in the bathroom with a bottle. |
| Generated sentence 3 | The man is holding a glass in the bathroom. |
| Generated sentence 4 | a person standing in the bathroom with a glass in his hand. |
| Generated sentence 5 | a man was standing in the bathroom holding a glass. |
| Generated sentence 6 | A person stands holding a mirror in the bathroom. |
| Generated sentence 7 | someone is standing in the bathroom holding a glass. |
| Generated sentence 8 | man stands in the bathroom with a glass. |
| Generated sentence 9 | a person is in the bathroom with a glass. |
| Generated sentence 10 | man standing in the bathroom and holding a glass. |

## 4. Experiments and Analysis

### 4.1. Dataset

Charades-STA is a benchmark dataset built upon the Charades dataset [44]. The Charades dataset is collected for action recognition as well as localization, including 9848 indoor videos of common daily actions. Temporal annotation and text descriptions are added to Charades by Gao [5], and the extended dataset is called Charades-STA. Charades-STA can be used in video captioning as well as video grounding. There are 6672 videos selected from Charades to Charades-STA, containing 16,128 moment–sentence pairs with 12,408 pairs in the training set and 3720 in the test set.

ActivityNet Captions is the largest dataset in this task, including 19, 209 indoor and outdoor videos collected from Youtube. Compared with Charades-STA, it is a much more complex dataset with 200 activity classes. The public dataset includes a training set (37,417 moment–sentence pairs), val1 set (17,505 moment–sentence pairs) and val2 set (17,031 moment–sentence pairs). We used Val 2 as the test set.

### 4.2. Experiment Settings

In this task, the evaluation metric is "R@n, IoU = m" [5], which is defined as the percentage of language queries having at least one "correct" instance of retrieval in the top-$n$ retrieved moments. A retrieved moment is "correct" when its IoU with the ground-truth moment is greater than $m$. Following the setting of previous methods, $n$ is $\{1, 5\}$, $m$ is $\{0.5, 0.7\}$ in Charades-STA and $\{0.3, 0.5\}$ in ActivityNet Captions.

The text encoder and image encoder of CLIP are used as feature extractors. As there are two models provided as image encoders in CLIP, the ViT model is adopted. The proposed algorithm is compared with several state-of-the-art methods from supervised, weakly supervised, and unsupervised perspectives.

The supervised methods not only use the video-grounding dataset but also pre-trained models to extract video features. The typical structure of the pre-trained model is C3D [45] or I3D [46]. Sports1M is usually used as a pretraining corpus with over 1 million videos. For supervised methods, different pre-trained models can influence performance greatly. Our method does not use any data in the video-grounding training sets and we only use the test sets to verify the performance. For all the listed methods, the data dependency rank is supervised methods, weakly supervised methods, unsupervised methods, and our zero-shot method. It is not fair to compare our method with the supervised methods, so we will mainly focus on weakly supervised methods and unsupervised methods.

### 4.3. Comparison with State-of-the-Art Methods

Results on Charades-STA are shown in Table 3. For supervised methods, CTRL [5], MAN [7], 2D-TAN [2] and MS-2D-TAN [8] are used. To display the best approaches for each metric, we used bolded numbers in the Tables. Supervised methods are compared together while the weakly supervised methods, unsupervised method and our zero-shot method are compared together.MS-2D-TAN [8] performs best of the supervised methods. For weakly supervised methods, TGA [14] and SCN [47] are chosen. DSCNet [16] is a method in the unsupervised manner. Compared with supervised methods, our method is better than CTRL [5] and MAN[7], while worse than 2D-TAN [2] and MS-2D-TAN [8]. Compared with the weakly supervised methods and unsupervised method, our zero-shot method has a significant improvement in "R@1 IoU = 0.5", "R@1 IoU = 0.7", and "R@5 IoU = 0.5". Comparing with TGA [14], all of the scores in our method outperform by a large margin, especially for "R@1", where our scores are nearly double. Though SCN [15] performs weakly in "R@1", it is good in "R@5". As we can see, the advantage of our method is more obvious in "R@1" than "R@5", and the reason may be both video and text are relatively simpler in this dataset. Our method is good at locating the most suitable atom action periods.

The comparative results of ActivityNet Captions are shown in Table 4. In the weakly supervised methods, only SCN is used, as TGA does not report the performance on Activi-

tyNet Captions. The performances of weakly supervised method SCN [15], unsupervised method DSCNet [16], and our method are very close. Our algorithm performs better in "R@1 IoU = 0.3" and "R@5 IoU = 0.3". However, as AZVG is not trained on certain video-grounding datasets, it has fewer domain limitations and wider applications.

**Table 3.** Performance of Charades-STA.

| Description | Method | R@1 IoU = 0.5 | R@1 IoU = 0.7 | R@5 IoU = 0.5 | R@5 IoU = 0.7 |
|---|---|---|---|---|---|
| Supervised | CTRL [5] | 23.63 | 8.89 | 58.92 | 29.52 |
| | MAN [7] | 41.24 | 20.54 | 83.21 | 51.85 |
| | 2D-TAN [2] | 39.81 | 23.25 | 79.33 | 52.15 |
| | MS-2D-TAN [8] | **60.08** | **37.39** | **89.06** | **59.17** |
| Weakly Supervised | TGA [14] | 19.94 | 8.84 | 65.52 | 33.51 |
| | SCN [15] | 23.58 | 9.97 | 71.80 | **38.87** |
| Unsupervised | DSCNet [16] | 28.73 | 14.67 | 70.68 | 35.19 |
| Zero-shot | Ours | **39.01** | **17.55** | **73.04** | 36.99 |

**Table 4.** Performance of ActivityNet Captions.

| Description | Method | R@1 IoU = 0.3 | R@1 IoU = 0.5 | R@5 IoU = 0.3 | R@5 IoU = 0.5 |
|---|---|---|---|---|---|
| Supervised | CTRL [5] | 47.43 | 29.01 | 75.32 | 59.17 |
| | CMIN [48] | **63.61** | 43.40 | 80.54 | 67.95 |
| | 2D-TAN [2] | 59.45 | 44.51 | 85.53 | 77.13 |
| | MS-2D-TAN [8] | 61.16 | **46.56** | **86.91** | **78.02** |
| Weakly Supervised | SCN [15] | 47.23 | **29.22** | 71.45 | 55.69 |
| Unsupervised | DSCNet [16] | 47.29 | 28.16 | 72.51 | **57.24** |
| Zero-shot | ours | **47.37** | 25.25 | **73.78** | 51.45 |

### 4.4. Ablation Study

4.4.1. Effectiveness of Bi-Directional Search in ATPD

In ATPD, a bi-directional search is an essential step. Different from shot boundary detection, our target is to obtain the target period related to a sentence, whereas shot boundary detection aims to detect the position of frames where the shot changes. Occasionally, the start and end moments of our target are exactly the boundaries of one shot, but not generally. Sometimes our target period is composed of several video shots. Using a bi-directional search, we can find consecutive shots related to the sentence.

Though atom regions in ATPD are similar to video shots that are selected by transition detection, our requirements for high-quality boundaries are relatively less rigorous. As the result is usually the combination of several atom regions, text information could be used as a guide to rectify some over-segmentation cases. In addition, the metric "R@n, IoU = m" is not sensitive to slight boundary changes. As a result, our atom region segmentation method could be replaced by any shot boundary detection method without obvious performance degradation due to the contribution of the bi-directional search module.

Tables 5 and 6 show the comparison of ATPD with and without bi-directional search in Charades-STA and ActivityNet Captions. The PSG module is not used in this ablation study. If there is no bi-directional search, the model will select the atom region containing the highest text–image similarity frame. As shown in the result, if the bi-directional part is removed, there will be a significant decrease in performance.

**Table 5.** Ablation study of bi-directional search for Charades-STA (without PSG).

| Bi-Directional Search | R@1 IoU = 0.5 | R@1 IoU = 0.7 | R@5 IoU = 0.5 | R@5 IoU = 0.7 |
|---|---|---|---|---|
| | 13.39 | 4.70 | 8.76 | 29.81 |
| ✓ | **37.01** | **16.85** | **72.72** | **36.85** |

**Table 6.** Ablation study of bi-directional search for ActivityNet Captions (without PSG).

| Bi-Directional Search | R@1 IoU = 0.3 | R@1 IoU = 0.5 | R@5 IoU = 0.3 | R@5 IoU = 0.5 |
|---|---|---|---|---|
| | 14.44 | 7.33 | 30.29 | 15.40 |
| ✓ | **46.88** | **25.07** | **70.89** | **46.99** |

### 4.4.2. Effectiveness of PSG

According to our observation, the original sentence is not always the best one to obtain the most accurate feedback via CLIP. The new generated prompting sentences share the same meaning as the original one with large diversity in the expression. Therefore, combining the new sentences together with the original one as "ensembling" will make the result more robust. This is similar to test time augmentation [49] in the image-classification task.

The purpose of designing the PSG module is to generate prompting sentences with "exactness" and "diversity". Two perspectives of PSG are proposed: the language-level method and the sentence-level method. The language-level method is usually more robust than the sentence-level method in alleviating wrong back-translation results, whereas the sentence-level method will generate more diverse results than the language-level method. In this part, ten prompting sentences will be generated for each original sentence using different methods.

To verify the effectiveness of "exactness" and "diversity" in PSG, we performed ablation studies under three situations: both "exactness" and "diversity", only "diversity" and only "exactness". In language-level PSG, only "diversity" means ignoring the semantic matching part and selecting languages of the highest diversity. Only "exactness" means selecting languages with the highest semantic matching scores. In sentence-level PSG, only "diversity" means for each sentence, the prompting sentences are selected by highest diversity. Only "exactness" means for each sentence, the prompting sentences are selected by the highest similarity scores. Results are shown in Tables 7 and 8 with middling fusion used. We can see that both "exactness" and "diversity" are helpful to improve performance, but "diversity" is more important than "exactness". If only considering "exactness", repeated sentences may be chosen. If only considering "diversity", sentences with the wrong meaning may be chosen. Our method can tolerate a small amount of wrong data. This phenomenon is consistent with our understanding of "ensembling". The results of random prompting sentence selection and no PSG are also listed for comparison. Considering only "diversity" generally performs better than random prompting sentence selection. Considering only "exactness" is generally weaker than random prompting sentence selection, but is better than no PSG.

Two ways of employing generated prompting sentences are also compared: averaging and middling. As shown in Tables 9 and 10, middling performs better than averaging, especially for the R@1 results. There are two main reasons: First, in the averaging fusion method, some incorrect sentences will influence the result, whereas in the middling fusion method, the effect of incorrect sentences will be alleviated. Secondly, as the top-1 peak position comes from the averaged new curve, sometimes it differs from any previous peaks in the frame–text similarity curve, which will cause the final peak position to be not the "real" local maximum.

**Table 7.** Eploring "exactness" and "diversity" of PSG using Charades-STA (middling fusion).

| PSG Type | Diversity | Exactness | R@1 IoU = 0.5 | R@1 IoU = 0.7 | R@5 IoU = 0.5 | R@5 IoU = 0.7 |
|---|---|---|---|---|---|---|
| Language-level | ✓ | ✓ | **39.01** | 17.55 | **73.04** | 36.99 |
| | ✓ | | 38.16 | 17.32 | 72.49 | **37.19** |
| | | ✓ | 37.42 | 16.94 | 72.74 | 36.85 |
| Sentence-level | ✓ | ✓ | 38.68 | **17.58** | 72.63 | 36.88 |
| | ✓ | | 38.44 | 17.23 | 72.34 | 37.04 |
| | | ✓ | 37.47 | 16.94 | 72.77 | 36.91 |
| Random selection | | | 37.98 | 17.45 | 72.66 | 36.99 |
| Without PSG | | | 37.01 | 16.85 | 72.72 | 36.85 |

**Table 8.** Eploring "exactness" and "diversity" of PSG using ActivityNet Captions (middling fusion).

| PSG Type | Diversity | Exactness | R@1 IoU = 0.3 | R@1 IoU = 0.5 | R@5 IoU = 0.3 | R@5 IoU = 0.5 |
|---|---|---|---|---|---|---|
| Language-level | ✓ | ✓ | **47.37** | 25.25 | **73.78** | 51.45 |
| | ✓ | | 47.18 | 25.25 | 73.04 | 50.98 |
| | | ✓ | 46.92 | 25.11 | 71.67 | 47.91 |
| Sentence-level | ✓ | ✓ | 47.21 | **25.43** | 73.69 | **51.58** |
| | ✓ | | 47.08 | 25.31 | 70.49 | 46.94 |
| | | ✓ | 47.11 | 25.18 | 70.33 | 46.57 |
| Random selection | | | 47.03 | 25.13 | 71.53 | 47.89 |
| Without PSG | | | 46.88 | 25.07 | 70.89 | 46.99 |

According to the results, the language-level PSG approach outperforms the sentence-level PSG method. Both methods are helpful compared with no PSG and random prompting sentence selection. In Charades-STA, PSG mainly improves the R@1 performance, and in ActivityNet Captions, both R@1 and R@5 results are enhanced. The performance of PSG illustrates the fact that efforts from the text end are useful in the video-grounding task. In the future, more experiments will be performed to use PSG in not only the zero-shot method but also some supervised, weakly supervised, and unsupervised methods in the decision process. Furthermore, PSG may be tried on other multi-modality tasks.

**Table 9.** Performance of PSG for Charades-STA.

| PSG Type | Fusion | R@1 IoU = 0.5 | R@1 IoU = 0.7 | R@5 IoU = 0.5 | R@5 IoU = 0.7 |
|---|---|---|---|---|---|
| Language-level | Middling | **39.01** | 17.55 | **73.04** | **36.99** |
| | Averaging | 38.31 | 17.12 | 72.47 | 36.29 |
| Sentence-level | Middling | 38.68 | **17.58** | 72.63 | 36.88 |
| | Averaging | 38.04 | 17.58 | 72.20 | 36.80 |
| Random selection | Middling | 37.98 | 17.45 | 72.66 | 36.99 |
| | Averaging | 37.80 | 17.39 | 72.71 | 36.64 |
| Without PSG | | 37.01 | 16.85 | 72.72 | 36.85 |

**Table 10.** Performance of PSG for ActivityNet Captions.

| PSG Type | Fusion | R@1 IoU = 0.3 | R@1 IoU = 0.5 | R@5 IoU = 0.3 | R@5 IoU = 0.5 |
|---|---|---|---|---|---|
| Language-level | Middling | **47.37** | 25.25 | **73.78** | 51.45 |
| | Averaging | 46.82 | 25.31 | 70.61 | 47.00 |
| Sentence-level | Middling | 47.21 | **25.43** | 73.69 | **51.58** |
| | Averaging | 47.10 | 25.32 | 71.03 | 47.31 |
| Random selection | Middling | 47.03 | 25.13 | 71.53 | 47.89 |
| | Averaging | 46.96 | 25.14 | 71.33 | 47.32 |
| Without PSG | | 46.88 | 25.07 | 70.89 | 46.99 |

*4.5. Discussion*

As shown in the experiment, AZVG shows competitive performance in a zero-shot manner in both Charades-STA and ActivityNet. The knowledge gained from the 400 million image–sentence pairs training corpus of CLIP has made the largest contribution to the zero-shot capability. Strictly speaking, this is not a rigorous zero shot that can generalize to completely new concepts. However, since the majority of common sense is covered by the knowledge in the video content, we can apply AZVG to any new activities without any more training. This is a very important feature for practical applications.

This work provides a new perspective to solve the video-grounding problem. There are two main advantages of this work over other existing methods: being training-free and able to be used in wider application scenarios. When encountering a new scenario, the existing methods need to collect data, make annotations, and train for a new network. The process is time-consuming and the annotation cost is high. It will be more difficult if you are faced with an abnormal scenario that lacks data. However, our method does not have those problems. As we do not need to train for every new scenario, it will be cheaper and more convenient for real applications. We can also cope with no-data scenarios.

There are still some limitations. First, as a large-scale pretrained model, CLIP is like a black box. We cannot control what concepts it is good at and what it is not good at. Maybe finetuning can solve this problem, but it is challenging to keep the balance of performance and generalization capability. Secondly, our atom action segmentation approach is based on the similarity of visual features, not semantics. Sometimes it is unstable when facing some abstract and complex scenes.

**5. Conclusions and Future work**

In this paper, we presented a brand-new atom-based zero-shot video grounding (AZVG) technique aiming for sustainable smart-city applications. AZVG is demonstrated using knowledge in large-scale pre-trained models such as CLIP, which can provide an image encoder and a text encoder. For the video-grounding task, AZVG achieves a good performance on the Charades-STA datasets and the ActivityNet Captions dataset, competitive with the weakly supervised learning methods and the unsupervised method. We do not need to train, which is a significant distinction between AZVG and other approaches. This reduces reliance on datasets and lowers labeling expenses. As a result, the zero-shot approach has a substantial potential advantage over previous training-based approaches in that it can be used in a wider range of urban living circumstances.

There are three future directions based on this work. First, further study of PSG. The effort from the text end seems to be very helpful. Currently, PSG is an independent module with only one sentence as the input. It is worth trying to use visual information to guide the generation of the best sentences. Secondly, finetuning-based solutions under specific usage scenarios. It is important to find a way to balance performance and generalization ability. Thirdly, the acceleration. In this method, we did not focus on the speed and there is still considerable room for improvement. If the customer wants to use a large-scale database, acceleration is necessary.

**Author Contributions:** Conceptualization, P.W., L.W. and J.S.; methodology, P.W. and L.S.; software, P.W.; validation, P.W. and L.S.; formal analysis, P.W.; investigation, J.S.; resources, L.W.; data curation, P.W. and L.S.; writing—original draft preparation, P.W.; writing—review and editing, J.S.; visualization, P.W.; supervision, L.W. and J.S.; project administration, J.S. All authors have read and agreed to the published version of the manuscript.

**Funding:** This research received no external funding.

**Institutional Review Board Statement:** Not applicable.

**Informed Consent Statement:** Not applicable.

**Data Availability Statement:** Not applicable.

**Conflicts of Interest:** The authors declare no conflict of interest.

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
