# Peer review of "Zero-Shot Video Grounding for Automatic Video Understanding in Sustainable Smart Cities"

_sustainability, doi:10.3390/su15010153_

Round 1

Reviewer 1 Report

In this manuscript, the author proposes a novel atom-based zero-shot video grounding (AZVG) method that divides the task into two sub-problems: First, they generate candidate time anchors that are independent of one another yet most closely related to the sentence. Secondly, to estimate concise periods for the input query, they segment and merge atom actions. Besides, a new augmentation concept of prompting sentences generation (PSG) is presented to achieve more robust performance. This research has definite innovation. However, I still have some comments for this.

1,” To get the concise moment, atom-based time period detection (ATPD) is proposed. This process includes two steps: splitting the video into atom actions and using bi-directional search to merge the anchor atom regions with surrounding regions under some rules.” What are those rules?

2, Should detail the” the time-interval determinantal point process (TI-DPP)”?

3,” The c + 1 sentences include the original sentence and c sentences generated by the language-level PSG.” What bothers me is whether the original sentence itself is the most accurate?

4,” Besides, a new way of evaluating the outlook similarity of two sentences is suggested.” But in section 3.3.1 semantic matching, I only find the MPNet pre-trained weights is chosen as a semantic matching filter. Is this a New way?

Author Response

Dear reviewer:

    We greatly appreciate your valuable comments. Please see the modified paper in the attachment.

Point 1: “To get the concise moment, atom-based time period detection (ATPD) is proposed. This process includes two steps: splitting the video into atom actions and using bi-directional search to merge the anchor atom regions with surrounding regions under some rules.” What are those rules?

Response 1: Thank you for your valuable comment. We considered three factors for each atom action period (the definitions are shown in Equation (7) – (10) on Page 10, Section 3.2): the peak value of the frame-text similarity curve (Equation (7)), the frame-text similarity flatness(Equation(8)), and the compactness(Equation(9)). The rules about how to use them are shown in Algorithm 1 (“Bi-directional search for the time period” in Page 10).

Point 2: Should detail the” the time-interval determinantal point process (TI-DPP)”?

Response 2: Thank you for your valuable comment. As introduced in Section 3.1 (please see page 7), TI-DPP is based on DPP (determinant point process), a mathematical model for subset selection (see reference [41]). We modified the DPP model to fit our scenario and set the optimization target: to find a subset of anchor frames with high text-image similarity scores and longer time intervals between each other. The description of the original DPP may not be sufficient. So we updated the manuscript by adding more descriptions of DPP in page 7.

Point 3: “The c + 1 sentences include the original sentence and c sentences generated by the language-level PSG.” What bothers me is whether the original sentence itself is the most accurate?

Response 3: Thank you for your valuable comment. The original sentence is not always the best one to get the most accurate feedback via CLIP. As shown in Table 1 (in Page 14), the prompting sentences have the same meaning as the original sentence, and the ground truth should be the same. However, for CLIP, its reactions to “a man stands in the bathroom holding a glass.” and “a man is standing in the bathroom with a glass” will be slightly different and we cannot control which one is better. Using the new sentences together with the original one as “ensembling” will make the result more robust. We updated the manuscript by adding such information in Section 4.4.2 “Effectiveness of PSG”.

Point 4: “Besides, a new way of evaluating the outlook similarity of two sentences is suggested.” But in section 3.3.1 semantic matching, I only find the MPNet pre-trained weights is chosen as a semantic matching filter. Is this a New way?

Response 4: Thank you for your valuable comment. “A new way” refers to the newly proposed metric “Word Set IoU” to evaluate the outlook similarity of two sentences (please see Section 3.3.2, Equation 11 on page 13). Comparing with the existing metrics in NLP, we reduce the impact of the words’ order in one sentence because CLIP is not so sensitive to the order of the words.

Reviewer 2 Report

Zero-shot Video Grounding for Automatic Video Understanding in Sustainable Smart City

The authors proposed a novel atom-based zero-shot video grounding (AZVG) method and a new augmentation concept of prompting sentences generation (PSG) for video understanding. The authors stated that the proposed method achieved decent performance. But we hope you will be able to address these issues listed below. I recommend making minor changes for re-review, as follows.

1.     Abstract should be shortened appropriately. Please reduce the content of the abstract and emphasize the merits of the proposed scheme.

2.     The abstract should end with a brief statement regarding the significance and impact of this paper.

3.     In the Introduction section, the research background and significance of this topic are less introduced, and it is recommended to make some supplements.

4.     The description of Figure 2 is extremely brief, and the author is advised to describe in detail the connections between the various components in Figure 2.

5.     In the discussion section, the authors should describe how the study's novelty has broader implications for the field. Then, discuss how the results have a significant impact on future research.

Author Response

Dear reviewer:

We greatly appreciate your valuable comments. The modified paper is in the attachment.

Point 1: Abstract should be shortened appropriately. Please reduce the content of the abstract and emphasize the merits of the proposed scheme.

Response 1: Thank you for your valuable comment. We updated the manuscript by making the abstract more streamlined to emphasize the merits of the method (please see Abstract on page 1).

Point 2: The abstract should end with a brief statement regarding the significance and impact of this paper.

Response 2: Thank you for your valuable comment. We updated the manuscript by adding descriptions of the significance and impact of this paper (please see Abstract on page 1).

Point 3: In the Introduction section, the research background and significance of this topic are less introduced, and it is recommended to make some supplements.

Response 3: Thank you for your valuable comment. We added two paragraphs in the introduction part. One is for the application of the video grounding task in the real life in page 2. The other is to emphasize the problem of existing methods that can be solved by our work in page 3.

Point 4: The description of Figure 2 is extremely brief, and the author is advised to describe in detail the connections between the various components in Figure 2.

Response 4: Thank you for your valuable comment. Figure 2 is the pipeline of our method while Section 3 focuses on the details of each module. We updated the manuscript by adding more details about the connections between the various components when introducing Figure 2. Besides, according to another reviewer’s opinion, Figure 2 is moved to Section 3. As a result, the enriched description of Figure 2 is put in the beginning of Section 3 in page 5.

Point 5: In the discussion section, the authors should describe how the study's novelty has broader implications for the field. Then, discuss how the results have a significant impact on future research.

Response 5: Thank you for your valuable comment. We updated the manuscript by adding the novelty and application potential of this work in the discussion section in page 19. Besides, in the conclusion part, we added a paragraph for future searches to introduce three possible directions based on the work in page 20.

Reviewer 3 Report

The authors presented zero-shot video grounding for automatic video understanding in smart cities. The proposed approach is divided into candidate anchors generation, Atom-based Time Period Detection (ATPD), and Prompting Sentences Generation (PSG). The proposed work is experimented using Charades-STA, and obtained results are promising. The idea discussed in this paper is very interesting.

I have the following recommendations to improve the paper:

- I suggest shifting figure 2 to section 3.

- I suggest adding the paper's organization at the end of the introduction.

- The authors have to add more details and descriptions about their approach. In addition, they should justify the use of these techniques.

- An important aspect is missing in this paper, which is what makes the proposed approach different from the other related works.

- to make this work reproducible, I believe sharing the code of the proposed method is highly appreciated.

- The discussion part should be extended to provide more insights about the paper and discuss the proposed study's limitations and strengths.

- Future works should be added to the conclusions to guide readers for possible extensions of this paper.

Author Response

Dear reviewer:

We greatly appreciate your valuable comments and we uploaded the modified paper. Please see the attachment.

Point 1: I suggest shifting figure 2 to section 3.

Response 1: Thank you for your valuable comment. We updated the manuscript by shifting figure 2 to section 3. We also added more introduction to the figure according to another reviewer’s opinion.

Point 2: I suggest adding the paper's organization at the end of the introduction.

Response 2: Thank you for your valuable comment. We updated the manuscript by adding the organization of this paper at the end of the introduction in page 3.

Point 3: The authors have to add more details and descriptions about their approach. In addition, they should justify the use of these techniques.

Response 3: Thank you for your valuable comment. We added more details of the proposed solution: a detailed description of the whole pipeline is added in page 5. A description of DPP is added in page 7. The effectiveness of PSG is analyzed in page 17. The novelties and limitations of the proposed method is discussed in page 19. The future directions are added in the conclusion in page 20. The effectiveness of the proposed method is shown in Table 3 and 4. Although our method is zero-shot based, the performance is on par with SOTA semi-supervised and unsupervised methods on ActivityNet dataset and much better than these methods on Charades-STA dataset.

Point 4: An important aspect is missing in this paper, which is what makes the proposed approach different from the other related works.

Response 4: Thank you for your valuable comment. The main advantage of our solution is the zero-shot capability. In the introduction section, we list the problem of training-based methods, no matter they are supervised, semi-supervised or unsupervised in page 3. We also added the merit of zero-shot-based method in the discussion section in page 19.

Point 5:  to make this work reproducible, I believe sharing the code of the proposed method is highly appreciated.

Response 5: Thank you for your valuable comment. We are going through the internal application for open source the proposed method and will share the code after getting the permission.

Point 6:  The discussion part should be extended to provide more insights about the paper and discuss the proposed study's limitations and strengths.

Response 6: Thank you for your valuable comment. We updated the manuscript by adding more limitations and strengths in the discussion part in page 19.

Point 7:  Future works should be added to the conclusions to guide readers for possible extensions of this paper.

Response 7: Thank you for your valuable comment. We updated the manuscript by adding a paragraph on Future works in the last section in page 20.

Round 2

Reviewer 1 Report

I suggest publishing the paper.

Reviewer 3 Report

The authors considered all my comments. I suggest accepting the paper in its current version.